# Synthetic Feed Attractants in European Seabass (*Dicentrarchus labrax*) Culture: Effects on Growth, Health, and Appetite Stimulation

**DOI:** 10.3390/ani15142060

**Published:** 2025-07-12

**Authors:** Federico Conti, Matteo Zarantoniello, Nico Cattaneo, Matteo Antonucci, Elena Antonia Belfiore, Ike Olivotto

**Affiliations:** 1Department of Life and Environmental Sciences, Università Politecnica delle Marche, Via Brecce Bianche 10, 60131 Ancona, Italy; n.cattaneo@pm.univpm.it (N.C.); e.a.belfiore@pm.univpm.it (E.A.B.); i.olivotto@univpm.it (I.O.); 2Department of Agricultural, Food and Environmental Sciences, Università Politecnica delle Marche, Via Brecce Bianche 10, 60131 Ancona, Italy; 3Independent Researcher, Via Pola 18, 64014 Martinsicuro, Italy; matteo.antonucci.89@outlook.com

**Keywords:** aquaculture, aquafeed, sustainability, animal welfare, histology

## Abstract

Improving feed palatability is a key challenge of aquaculture to reduce unconsumed feed and preserve water quality. The introduction of standardized and cost-effective processes to produce synthetic flavors as attractants offers a sustainable solution to minimize environmental impact. In this context, the present study aimed to investigate the potential role of synthetic flavors, previously tested in zebrafish, as feed attractants in European seabass (*Dicentrarchus labrax*), a carnivorous and commercially relevant species for Mediterranean aquaculture. Using a multidisciplinary approach—combining biometric measurements with histological and molecular analyses—this study focused on fish zootechnical performance, welfare, and central appetite regulation pathways. The diet supplemented with a caramel-synthetic flavor led to increased feed intake, which subsequently enhanced fish growth performance, confirming its attractiveness to this species. Moreover, analysis of central molecular pathways revealed that the caramel flavor was the most effective among those tested in stimulating appetite and feed consumption. Assessing synthetic flavors as feed attractants for commercially important species is essential for optimizing feeding strategies and nutritional management in aquaculture, contributing to more efficient and sustainable production systems.

## 1. Introduction

In a blue economy context, the aquafeed industry is achieving considerable progress in enhancing resource utilization [1,2]. In fact, the use of fish meal and oil for diet formulations, despite being ideal ingredients, is no longer economically and environmentally sustainable due to their increasing cost and the volume of wild-caught forage fish needed [3,4,5]. Therefore, the aquafeed production sector has been driven to explore several alternative ingredients or feed additives [6,7,8,9]. However, a high inclusion of novel protein sources, especially plant-based ingredients, can negatively affect the palatability of aquafeeds [10,11,12,13]. In fact, the off-flavor substances present in these ingredients can affect the feed’s attractiveness, resulting in a reduced feed intake and thus compromising the growth performances of farmed fish [14,15]. Moreover, the resulting increase in uneaten feed can have long-term impacts on both farm economics and on the aquatic environment [16,17]. Improving feed palatability is therefore a key challenge, especially for carnivorous fish fed diets with reduced levels of marine-derived ingredients [18,19,20,21].

In this context, feed attractants can be included in aquafeed formulations to enhance flavor and stimulate the feed intake in fish [22,23]. Molecules released by the feed attractants are received by the fish olfactory and gustatory systems, which in turn activate the central nervous system and drive feeding behaviors [23,24]. In addition, this sensory stimulation can trigger gastric secretion, promoting digestion and nutrient absorption through the release of peptides and hormones [25]. In fact, the brain is the key site that receives and integrates information about the nutritional state and energy level of the organism, and central-homeostatic signals (e.g., Neuropeptide Y) typically regulate the nutrient intake [26,27]. In addition, in fish, monoaminergic systems are part of the neural network that controls feeding, and both serotonin (or 5-hydroxytryptamine) and dopamine receptors are directly involved [28]. In this context, the hedonic properties of feed—driven by its palatability—can influence these regulatory mechanisms by activating the brain’s reward system in fish, thereby stimulating feed intake [29,30].

Numerous studies have demonstrated that some specific compounds, like L-amino acids, small chain peptides, betaine, and nucleotides, are potent feed attractants for fish [31,32]. In particular, it has been shown that the nucleotides found in the muscle extract of jack mackerel produced the strongest feeding stimulatory response in greater amberjack (*Seriola dumerili*) [33]. These compounds are naturally abundant in marine by-products—such as meals and extracts from forage fish, mollusks, squids, and shrimps—which has led to their inclusion in aquafeeds as effective natural attractants [34,35]. However, the reliance on marine-derived resources raises sustainability concerns [36], especially since their attractant efficacy varies depending on raw material origin, quality, freshness, and processing methods [3,37]. To address these aspects, alternative molecules are currently being used to reduce the dependence on marine ingredients. In this context, Yesilayer and Kaymak [18] found that 1% crystalline betaine supplementation in a diet rich in fish meal and oil (with 25% soybean meal) positively influenced growth performance and feed utilization in rainbow trout (*Oncorhynchus mykiss*) juveniles. Differently, amino acids, despite being recognized as the major class of compounds with an attractive effect, present several limitations due to their concentration and interaction with other dietary components as well as their different effectiveness in different fish species and life cycle stages [38,39]. Hence, when these molecules are singly provided, they are generally considered less effective in attracting fish compared to the extracts of marine-derived raw materials [32,38].

Synthetic flavors derived from standardized processes have recently been proposed as novel and promising alternatives to the traditional feed attractants [40,41]. In fact, these compounds are more accessible than natural attractants since they can be produced even if they are not found in a particular region or are unavailable during certain harvesting periods. Additionally, synthesizing the desired substances is more cost-effective than extracting them directly from raw materials, as it requires fewer steps and reagents to obtain the final product, simplifying the production processes and reducing costs [42]. Moreover, synthetic flavors can be used in their pure form and at precisely the required concentration, ensuring a consistent quality and composition while providing standardized effects during feeding [40,42].

Preliminary studies on the entire life cycle of zebrafish (*Danio rerio*) showed that dietary supplementation with 1% (*w*/*w*) attractive synthetic flavors positively influenced appetite stimulation, feed intake, and, as a result, fish growth [40,41]. Zebrafish has been recognized as an ideal model for finfish nutrition studies relevant to the aquaculture industry [43,44], and findings from this species can serve as a valuable foundation for further research in farmed fish. While zebrafish is omnivorous, European seabass is a strictly carnivorous species with a high market value and of particular importance in the aquaculture sector of the Mediterranean region [45]. However, both species share evolutionarily conserved sensory mechanisms, including similar olfactory receptor families, signal transduction pathways (e.g., appetite-related hormones), and neural circuitry (e.g., dopamine receptors), suggesting that similar feeding cues can promote feed intake across different dietary types [46,47,48]. Despite their varied nutritional requirements, both groups respond to common sensory signals that trigger appetite, enabling the development of broadly applicable strategies to enhance feeding efficiency in aquaculture. This shared physiological basis supports the use of common feed attractants within zebrafish and carnivorous species like European seabass [49].

In this context, the present study aimed to assess the effectiveness of the synthetic flavors previously evaluated in zebrafish [40,41], in promoting appetite stimulus and enhancing the feed intake and growth performance in European seabass (*Dicentrarchus labrax*) during a 90-day feeding trial, along with evaluating the physiological responses related to fish health. Using a multidisciplinary laboratory approach that included biometric measurements, histological analysis, and molecular investigations—essential when testing new aquafeed ingredients or additives—this study focused on fish zootechnical performance, welfare, and the regulation of brain appetite systems. Production rates of European seabass are rising to meet the increasing seafood demand, with feed costs representing the largest portion of production expenses [50]. Incorporating synthetic flavors as feed attractants can help reduce feed waste while maintaining growth performance and productivity by enhancing diet palatability. Moreover, this shift toward more sustainable and innovative aquafeed formulations supports the adoption of eco-friendly practices within the aquaculture industry.

## 2. Materials and Methods

### 2.1. Ethics

The feeding trial and all the experimental procedures involving animals were conducted according to EU legal frameworks relating to the protection of animals used for scientific purposes (Directive 2010/63/EU). This study was approved by the Ethics Committee of Marche Polytechnic University (Ancona, Italy, n° 2; 1 December 2022) and the Italian Ministry of Health (Aut N° 453/2023-PR). Fish suffering was minimized by using the MS222 anesthetic (1 g/L; Merck KGaA, Darmstad, Germany).

### 2.2. Synthetic Flavors and Production of Experimental Diets

The two attractive (F25, cheese; F35, caramel) and one repulsive (F32-, coconut) flavors were produced by To Be Pharma S.r.l. (S. Egidio alla Vibrata, Teramo, Italy) following the same composition reported in Conti et al. [41], in compliance with the current sector-specific legislation, specifically Regulation (EC) Nos. 1334/2008 and 1333/2008. Briefly, each synthetic flavor consisted of a liquid blend of mixed aromatic chemicals added to 1,2-propanediol (propylene glycol; PG) as a functional solvent. F25 (cheese odor) contained trimethylamine, 2-acetylpyrazine, 2-acetylpyridine, and dimethyl sulfide; F35 (caramel odor) included vanillin, maltol, cyclotene, acetoin, butyric acid, and capric acid, with traces of γ-octalactone and γ-hexalactone; F32- (coconut odor) comprised γ-heptalactone, γ-nonalactone, δ-hexalactone, and vanillin, with traces of δ-octalactone and maltol. Due to intellectual property restrictions, compounds are listed in order of relative abundance. All the substances used are recognized as safe for use by the Flavor and Extracts Manufacturers Association and the Food and Drug Administration Agency for Toxic Substances and Disease Registry 2007. The safety of the flavors for both humans and animals has been confirmed through a technical data sheet and a Material Safety Data Sheet (MSDS), ensuring we respected the regulatory standards and safety for consumption.

A commercial diet for European seabass (Ecoprime, Veronesi, VR, Italy), rich in land animal proteins (LAPs), was used as a control diet (CTRL). For the available details of the control diet, please see https://fishfeed.eu/IT/alimentazione-pesce/mangime-Spigola/9/8/2/Spigola-ECOPRIME-3.aspx (accessed on 02 September 2023) and Table 1. Then, four experimental diets were prepared, starting from the commercial CTRL diet, by adding 1% (*w*/*w*) of PG, F25 flavor (attractive), F35 flavor (attractive), or F32- flavor (repulsive) (named PG, F25, F35, and F32- diets, respectively). The commercial diet was selected to replicate real farming conditions and minimize experimental variability. The PG and all the flavors were daily added at 1% (*w*/*w*) to the daily feed ration using a micropipette. The incorporation rate of 1% *w*/*w* was selected based on previous studies and aligned with concentrations used for synthetic flavors [40,41,51]. Subsequently, the mixture was vigorously mixed to ensure a uniform flavor distribution in the feed. This process was implemented to guarantee the consistency of the experimental conditions.

### 2.3. Experimental Design and Zootechnical Parameters

Five hundred and forty European seabass juveniles (initial body weight: 72.48 ± 8.04 g) provided by Panittica Pugliese (Torre Canne di Fasano, Brindisi, Italy) were acclimated for two weeks in a single 1000 L tank with mechanical and biological filters and a UV lamp (Panaque s.r.l., Viterbo, Italy), which ensured the following water parameters: temperature 21.0 ± 0.5 °C; salinity 30 ± 0.5 ‰; ammonia and nitrite < 0.05 mg/L; nitrate < 10 mg/L. During this period, fish were fed the CTRL diet (in agreement with the feed company instructions in relation to fish size and water temperature). At the end of acclimation period, fish were randomly divided into six experimental groups (in triplicate) as follows: (i) CTRL group, fish fed the CTRL diet; (ii) PG group, fish fed PG diet; (iii) F25 group, fish fed F25 diet; (iv) F35 group, fish fed F35 diet; (v) F32- group, fish fed F32- diet; and (vi) ROT group, fish fed the two attractive diets (F25 and F35), each administered singularly in a weekly rotation scheme, in order to avoid possible olfactory receptor adaptation in response to long-term stimulation [52,53,54]. Each experimental group (90 fish per group; 30 fish per tank) consisted of three 500L tanks provided with mechanical and biological filtration (Panaque s.r.l., Viterbo, Italy), ensuring the same water parameters of the acclimatation period.

The duration of the feeding trial was 90 days, during which the feed consumption was monitored daily, providing a pre-weighed quantity of feed corresponding to 1.5% of fish body weight (in agreement with the feed company instructions in relation to fish size and water temperature). A fixed quantity of feed was chosen to better reflect common farming practices for this species [55] and in accordance with Turchini and Hardy [56]. The quantity of feed was adjusted every 14 days by weighing all the specimens from each tank. Seabass were hand-fed the experimental diets, and the uneaten feed was recovered daily 15 min post-administration by siphoning, dried overnight in an oven at 40 °C, and weighed. The daily feed intake data were calculated by subtracting uneaten feed from the feed administered and then expressed as an average percentage of daily feed ingested. The 15 min duration was chosen based on the maximum duration of the behavioural preference test performed during flavor selection, as described in the study by Conti et al. [41], along with the fact that an effective feed attractant has to stimulate a rapid ingestion of the feed within the first minutes of administration [57]. This was considered as a methodological decision.

During the whole trial, a visual inspection of tanks was carried out daily to check for the presence of dead specimens.

At the end of the feeding trial, after 24 h of fasting, all the fish were euthanized with a lethal dose of MS222 (1 g/L, Merck KGaA) and then individually weighed.

Data were used to calculate the following:weight gain, WG (g) = final body weight − initial body weight(1)feed conversion ratio, FCR = total weight of feed consumed/(final body weight − initial body weight)(2)relative growth rate: RGR (%) = [(final body weight − initial body weight)/initial body weight] × 100(3)specific growth rate: SGR (%) = [(ln final body weight − ln initial body weight)/time (days)] × 100(4)

The survival rate was calculated as follows:SR (%) = (final number of fish/initial number of fish) × 100(5)

Finally, samples from the brain, liver, and distal intestine were collected and properly stored for further analyses, as reported in the sections below.

### 2.4. Histological Analysis

Samples of liver and distal intestine were collected from 10 randomly and blindly selected fish per tank (*n* = 30) and fixed for 24 h at 4 °C by immersion in Bouin’s solution (Merck KGaA, Darmstad, Germany). Samples, after being dehydrated in ethanol solutions (80, 95, 100%), washed with xylene (Bio-Optica, Milan, Italy), and preserved in paraffin (Bio Optica), were cut with a microtome (Leica RM2125 RTS, Nussloch, Germany) to obtain sections of 5 μm in thickness. Sections were stained with Mayer hematoxylin and eosin Y (EE; Merck KGaA) or Periodic Acid Schiff (PAS; Bio-Optica, Milan, Italy) and were then observed using a Zeiss Axio Imager.A2 microscope (Zeiss, Oberkochen, Germany). Images were acquired using a combined color digital camera (Axiocam 105, Zeiss), and measurements were taken with the ZEN 2.3 software (Zeiss). For each sample, the morphometric and histopathological evaluations were performed on three transversal sections collected at intervals of 200 μm. The histopathological scoring criteria measured for both distal intestine and liver sections were considered, as described both in Zarantoniello et al. [58] and Table 2. Specifically, for the distal intestine, the mucosal fold height (all the undamaged and non-oblique folds were considered) and submucosa width were measured. In addition, the events of mucosal fold fusion, the incidence of inflammatory influx in the submucosa, and the presence of supranuclear vacuoles were considered during the inspection of each section. As regards the liver, EE staining was used to assess hepatocyte morphology, the presence of pathological alterations, and the degree of fat accumulation, while the PAS staining allowed us to highlight glycogen deposition in hepatocytes. The fat fraction percentage was quantified on areas with no blood vessels or bile ducts using ImageJ software (ver. 1.54d) (setting a homogeneous threshold value) and expressed as a percentage of the area occupied by fat on the total hepatic parenchyma.

### 2.5. Molecular Analyses

Total RNA was extracted from samples of the whole brain and distal intestine of 5 fish per tank (15 per experimental group) using TRIzol^TM^ reagent (Invitrogen, Waltham, MA, USA) according to Cionna et al. [59]. Then, after checking the final RNA concentration (Nanophotometer P-Class, Implen, München, Germany) and the integrity, the complementary DNA was synthesized from 1 μg of total RNA using the iScript™ cDNA synthesis kit (Bio-Rad, Hercules, CA, USA).

The relative mRNA abundance of genes involved in fish appetite regulation (Neuropeptide Y, *npy*), the brain reward system (dopamine receptor D3, *drd3*), serotonergic system (5-hydroxytryptamine receptor 1A, *5ht1a*), and the immune response (interleukin 1β, *il1b*; interleukin 10, *il10*; tumor necrosis factor alpha, *tnfa*) was investigated on brain and intestine samples, respectively. As reference genes, beta-actin (*β-actin*) and 18S ribosomal protein (*18s*) were both used. The stability of both reference genes was evaluated across all experimental groups, and their geometric mean was used for normalization of target gene expression levels, in accordance with MIQE guidelines. The primer sequences used in the present study are reported in Table 3. The real-time quantitative PCR was performed in an iQ5 iCycler thermal cycler (Bio-Rad, Hercules, CA, USA) following the protocol described in Randazzo et al. [60]. Annealing temperatures for each primer were optimized using temperature gradient assays, and primer specificity was confirmed by the absence of primer–dimer formation and analysis of dissociation curves. Additionally, primer efficiencies were evaluated using a dilution series of cDNA from the CTRL group at concentrations of 1:1, 1:10, 1:100, and 1:1000. All primers tested exhibited efficiencies around 90%, with R^2^ values ranging from 0.995 to 0.998. At the end of each cycle, the fluorescent was monitored, and the melting curve presented a single peak for every product. Two no template controls were added for each reaction to guarantee the absence of contamination. Amplification products were sequenced, and homology was verified. The qPCR data were processed using the iQ5 optical system software version 2.0 (Bio-Rad), including GeneEx Macro iQ5 Conversion and genex Macro iQ5 files. Gene transcript expression variations among experimental groups are reported as relative mRNA abundance (arbitrary units).

### 2.6. Statistical Analysis

The tanks were used as the experimental unit for data related to the survival rate and zootechnical performance (*n* = 3), while fish were considered the experimental unit for all the remaining analyses (*n* = 15 and *n* = 30 for molecular and histological analyses, respectively). After being checked for normality (Shapiro–Wilk test) and homoscedasticity (Levene’s test), data from each analysis were analyzed using one-way analysis of variance (ANOVA) followed by Tukey’s multiple comparison post hoc test (software package Prism 8, Graphpad Software version 8.0.2, San Diego, CA, USA). Significance was set at *p* < 0.05.

## 3. Results

### 3.1. Zootechnical Performance

Table 4 reports zootechnical indices of European seabass fed the experimental diets. The survival rate was 100% in all the experimental groups. Considering growth performance, fish fed F35 were characterized by significantly (*p* < 0.05) higher values of final body weight, weight gain, and relative and specific growth rates compared to the CTRL, PG, F25, and F32- groups, which did not show significant differences among them. In the ROT group, all these indexes were lower compared to those observed in the F35 group, without statistically significant differences. In addition, when considering the feed conversion ratio, although ANOVA indicated significant differences (*p* = 0.0359), multiple comparison tests did not show this significance between any specific groups. Therefore, no pairwise significant differences were detected in FCR among experimental groups. Finally, as regards the daily feed intake, fish fed the F35 and ROT diets showed a significantly (*p* < 0.05) higher percentage of feed ingested with respect to all the other groups (with the exception of F25), which did not show significant differences among them.

### 3.2. Histological Analyses

In the distal intestine (Figure 1), no morphological or pathological alterations were observed in any of the experimental groups. Specifically, although significant differences were observed among the experimental groups in terms of both mucosal fold height and submucosa width (*p* < 0.05; Table 5), these differences did not represent pathological alterations. In fact, all measured values fall within the established physiological range for seabass [61]. Furthermore, distal intestine samples from all the experimental groups highlighted a physiological condition, with a regular columnar epithelium with polarized and basally located nuclei, and with a well-preserved architecture and no visible tissue damage or inflammation. Moreover, samples showed scarce lymphocyte infiltration and an absence of supranuclear vacuoles, and episodes of mucosal fold fusion were isolated.

As regards the liver, fish from all the experimental groups exhibited a physiological structure of the hepatic parenchyma (Figure 2). However, liver sections revealed a variable degree of fat accumulation among the experimental groups. Specifically, the F35 and ROT groups exhibited a significantly higher percentage of fat fraction compared to all the other experimental groups (Table 3), with hepatocytes characterized by an abundant intracytoplasmic accumulation of fat. Finally, no differences were evident among the experimental groups in terms of glycogen deposition, classified as modest in each PAS-stained section (Figure 2c).

### 3.3. Real-Time qPCR Results

*Appetite.* When considering *npy* relative gene expression (Figure 3), the F35 group showed a significant (*p* < 0.05) downregulation compared to the other experimental groups, which did not show significant differences among them.

*Monoaminergic system.* When considering the *5ht1a* relative gene expression (Figure 4a), the F35 and ROT groups showed a significant (*p* < 0.05) upregulation compared to both the CTRL and F32- groups, while the F25 and F32- groups did not show significant differences compared to the CTRL group. No significant differences were detected between the CTRL and PG groups.

Considering *drd3* relative gene expression (Figure 4b), the F32- and ROT groups showed a significant (*p* < 0.05) upregulation compared to the other experimental groups, which did not show significant differences among them.

*Immune response.* Considering the expression of markers involved in the immune response (Figure 5), groups fed the F35 and F32- diets showed a significant (*p* < 0.05) upregulation of *il1b*, *il10*, and *tnfa* compared to the other groups, which did not show significant differences among them.

## 4. Discussion

Assessing the effectiveness of synthetic flavors as feed attractants in diets for commercially important carnivorous fish species is crucial for enhancing feeding practices in aquaculture, especially given the limitations of current aquafeed formulations. The synthetic flavors used in the present study were previously tested in studies conducted on zebrafish considering its whole life cycle [40,41], proving their safety for fish and, for those that are effective attractants, their ability to stimulate the appetite, increase the feed intake, and promote growth. Given that zebrafish is a well-established model organism for nutrition studies [44] and, unlike most farmed fish species in the Mediterranean region, is omnivorous [62], this study aimed to evaluate the suitability of the same synthetic flavors on a commercially important carnivorous species, the European seabass. The conservation of appetite stimulation mechanisms between omnivorous and carnivorous fish suggests that similar feeding cues and attractants can effectively enhance feed intake across different dietary types. Despite their distinct nutritional needs, both omnivorous and carnivorous species respond to common sensory signals that trigger the appetite, allowing for the development of broadly applicable strategies to improve the feeding efficiency in aquaculture. This shared physiological basis supports the use of common feed attractants in both zebrafish and European seabass [49].

As a first consideration, the overall health status of target organs directly involved in feed processing must be analyzed when novel additives are included in the feed formulation. In the present study, the PG and all the tested synthetic flavors had no adverse effects on the liver and distal intestine, as widely demonstrated by the histological analysis. However, it is important to note that fish fed diets containing caramel and coconut flavors showed an upregulation of immune-related markers (*il10*, *il1b*, *tnfa*) in the distal intestine, despite the clear absence of inflammatory signs in histological analyses. Pro-inflammatory cytokines play a key role in initiating the inflammatory response by stimulating the proliferation of T and B lymphocytes as well as macrophages, which facilitates the migration of immune cells to the inflammation site [63,64]. In addition to immune functions, fish interleukins are also involved in regulating other physiological processes such as muscle growth and metabolism [65], adding complexity to the interpretation of these findings. Artificial dietary additives, including synthetic flavors, may influence fish immune responses by modulating gut microbiota [66]. However, it remains difficult to generalize beneficial or negative effects of synthetic flavors on fish gut health due to limited data [42]. In this study, the absence of morphological changes and moderate gene upregulation likely reflect a physiological adaptation rather than pathology. Nevertheless, future research should further investigate the interactions between dietary additives, gut microbiota, and immune function in aquaculture species. Moreover, the diverse roles of interleukins may explain the discrepancies observed in this study, highlighting the need for further research to clarify the less obvious relationship between synthetic flavor supplementation and fish cytokine gene expression. Finally, the absence of negative effects from the basic solvent was clearly demonstrated by the comparable results observed between fish fed the CTRL and PG diets.

Alongside maintaining optimal fish welfare, the primary goal of adding feed attractants to the diet is to enhance palatability, thereby supporting high production rates. In this study, fish fed a commercial diet supplemented either with the caramel synthetic flavor alone (F35 group) or in rotation with the cheese flavor (ROT group) exhibited a significantly higher daily feed intake compared to those fed the CTRL, PG, and coconut diets. This confirms the attractive effectiveness of the caramel flavor in this species, consistent with previous findings in zebrafish [40,41]. The increased daily feed intake observed in these groups throughout the feeding trial likely had a direct impact on the energy available to the fish for growth and nutrient storage. Specifically, both the caramel and ROT groups showed the highest weight gain and growth rates—although the ROT group’s results were not statistically different from the other groups—and showed a significantly higher liver fat content, which is a primary site of lipid storage in several marine fish species, including European seabass. Although lipid storage may reflect high energy availability, excessive hepatic lipid deposition can affect fish health over time, particularly under stress conditions or during energetic-demanding phases such as reproduction [67]. Therefore, while no acute health issues were identified during the trial, further research is warranted to evaluate the long-term consequences of the observed increased hepatic fat accumulation in fish. At any rate, this finding is consistent with a previous study on rainbow trout [68] that showed how the incorporation of feed attractants in a low palatable diet (based on potato protein concentrate) was able to enhance the fish feed intake, ensuring optimal growth. Additionally, in this study, no significant differences in feed conversion ratio (FCR) were observed, as all experimental groups received the same base diet. Therefore, the increased growth detected in groups fed diets supplemented with attractive flavors can be attributed solely to the higher feed intake.

The central-brain regulation of feeding in fish involves a fine-tuned cross-talk between homeostatic and hedonic pathways, both related to feed intake [27,28,69]. More specifically, Neuropeptide Y (a strong orexigenic peptide) and monoaminergic signals (e.g., serotonin and dopamine) are interconnected in the regulation of appetite and the energy balance, determining the final feeding response. In this study, the lower *npy* expression observed in the F35 (caramel) group reflected the higher feed ingestion observed. In fact, since *npy* expression typically increases during fasting, this inverse pattern suggests a higher nutrient availability, and thereby a higher satiation of the fish fed the caramel diet [70,71]. This result was consistent with findings observed in a previous study on adult zebrafish [40]. Along with the homeostatic signals, the hedonic properties of feed, including smell and taste, are known to activate pleasure sensations during feed ingestion, involving additional signals at the central level [24]. In teleost, the monoaminergic systems, which include serotonin and dopamine receptors, play a direct role in feeding regulation, inducing the activation of the brain reward system, which may increase the feed intake [28,29,30]. In this study, the higher expression of the serotonin 1A receptor (*5ht1a*) in both the caramel and ROT groups, compared to the CTRL and coconut groups, was consistent with *npy* expression and aligned with the observed increases in feed intake and fat accumulation. This suggests a state of greater satiety, likely resulting from greater energy reserve storage [72,73]. In contrast, the significant downregulation of the dopamine receptor *drd3* in the caramel group, compared to the ROT group, reaching levels similar to those in fish fed inert diets (CTRL and PG) and the group receiving the cheese flavor, may be associated with the reinforcement effects of prolonged exposure to a positive stimulus linked to reward mechanisms [74]. This could indicate reduced dopaminergic sensitivity following repeated stimulation, consistent with mechanisms of reward habituation described in vertebrates, although these mechanisms are still poorly investigated in fish [74,75,76]. These findings, alongside the reduced expression of the orexigenic signal *npy*, suggest a dual modulation of both homeostatic and hedonic feeding pathways. The consistent consumption of a highly palatable diet may have provided adequate nutritional availability, diminishing the need for NPY-driven hunger signals and leading to adaptations in the reward pathways [27,40,69,74]. This combined regulation suggests that fish in the caramel group likely experienced a prolonged state of satiation and reduced hedonic-driven feeding due to extended exposure to a positive stimulus. In contrast, for the ROT and coconut groups, it can be concluded that the continuous rotation scheme or the prolonged exposure to a potentially weakly perceived stimulus, respectively, was not sufficient to elicit consistent feeding stimulation. These findings underscore the importance of understanding the neurobiological mechanisms that drive feeding behavior and how different stimuli affect the feed intake and energy reserves. Future studies should consider conducting preference tests, such as multiple-choice feeding trials, which would help to provide more precise insights into flavor preference and feeding behavior in this species. Overall, the results of this study demonstrated that dietary supplementation with caramel flavor enhanced diet palatability, leading to an increased feed intake and improved growth performance in European seabass. This supports the role of attractive sensory stimuli in feed as key regulators of feeding motivation through central nervous system signals. In contrast, the less effective results observed in the ROT group are likely due to the inadequacy of the rotation scheme or the alternating use of caramel flavor with a less potent attractant (cheese), as fish in the cheese group consistently showed results comparable to the CTRL group. These findings highlight that results observed in carnivorous species like European seabass can be similar to those observed in omnivorous species, aligning with the preferences reported in zebrafish by Conti et al. [40,41]. Finally, the flavor previously identified as repulsive (coconut) did not act as a strongly negative stimulus in this fish species. Instead, it appeared to be a weakly perceived stimulus that did not significantly affect the overall acceptance of the basal diet, resulting in a zootechnical performance comparable to that of fish fed both the CTRL and PG diets.

## 5. Conclusions

All the synthetic flavors tested showed no effect on the overall health status of key metabolic organs, such as the liver and intestine, of the fish. The provision of a diet containing the F35 flavor (caramel odor) promoted a higher feed intake, improving fish growth performance, which confirmed the attractive role of this flavor even in European seabass. In addition, the modulation of both appetite and monoaminergic systems suggested that the addition of caramel flavor in the diet was the most valuable solution compared to other ones tested, improving feed palatability and the feeding response. While this study focused on physiological responses, the impact of dietary attractants on the sensory quality of fish flesh was not assessed. Given the importance of product taste in market acceptance, future studies should include sensory evaluations to determine whether flavored additives influence the final taste profile of the fish. Additionally, as a further step forward from the present study, future research should consider a faster, easier, and more precise delivery of the feed attractants to the feed. In conclusion, the present study demonstrates that carnivorous species such as European seabass share common sensory mechanisms for appetite stimulation with omnivorous species like zebrafish, facilitating the development of broadly applicable strategies to enhance the feeding efficiency in aquaculture.

## Figures and Tables

**Figure 1 animals-15-02060-f001:**
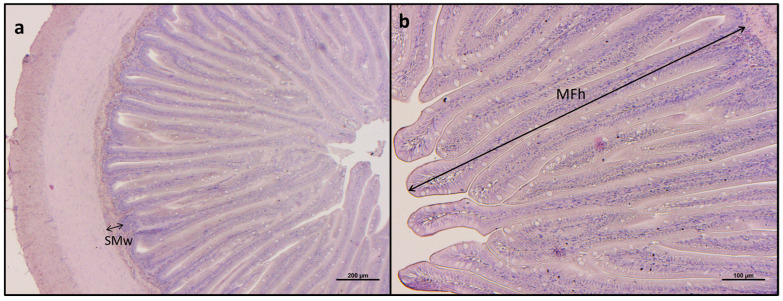
**Histological evaluation of intestine.** (**a**) Example of histomorphology of distal intestine from European seabass fed the F35 diet with measurement criteria for submucosa width (SMw); (**b**) example of histomorphology of distal intestine from European seabass fed the PG diet with measurement criteria for the mucosal fold height (MFh). Scale bars: (**a**) 200 µm; (**b**) 100 µm.

**Figure 2 animals-15-02060-f002:**
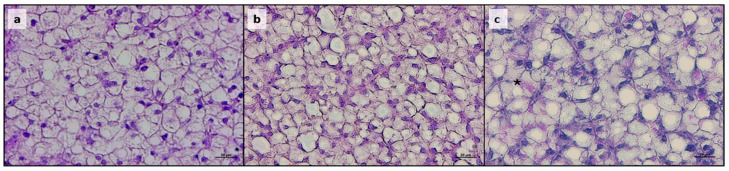
**Histological evaluation of liver.** Example of histomorphology of liver from European seabass fed the experimental diets. (**a**) CTRL; (**b**) F35; (**c**) ROT. Staining: (**a**,**b**) EE; (**c**) PAS, asterisk indicates glycogen deposition. Scale bars: 20 µm.

**Figure 3 animals-15-02060-f003:**
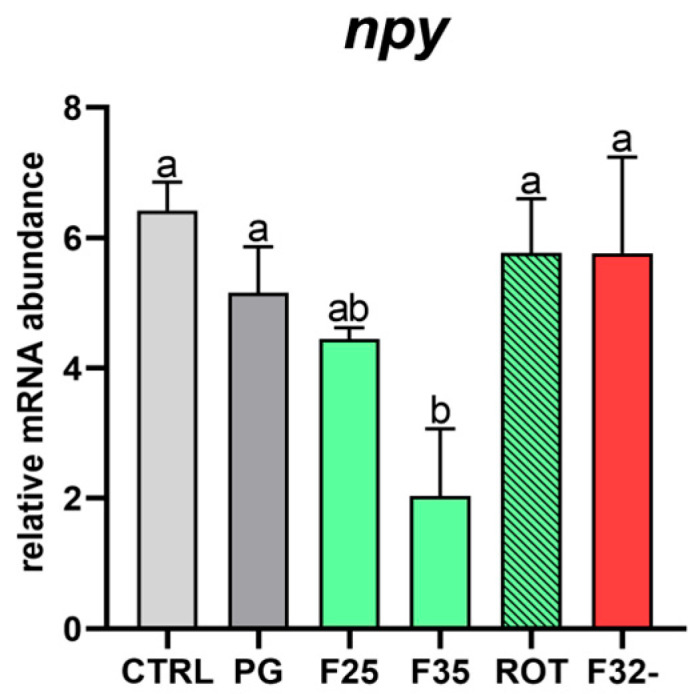
**Relative mRNA abundance of marker related to appetite regulation.** Real-time qPCR was performed on brain samples of European seabass fed the experimental diets. Results are ex-pressed as mean ± SD (*n* = 5). ^a,b^ Different letters denote statistically significant differences among the experimental groups.

**Figure 4 animals-15-02060-f004:**
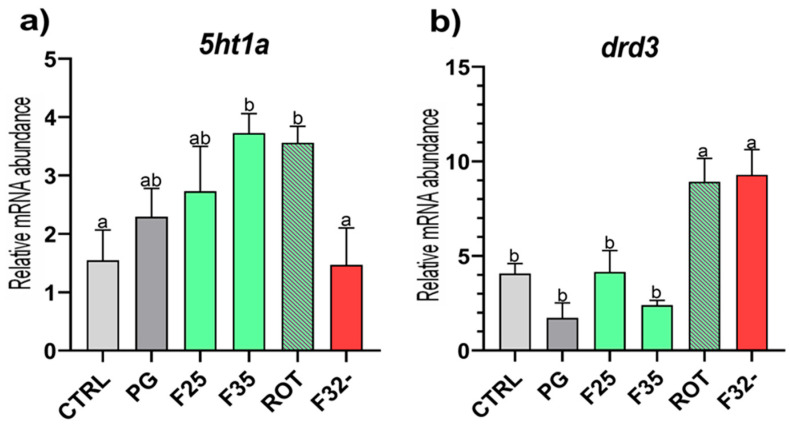
**Relative mRNA abundance of markers related to monoaminergic systems.** Real-time qPCR was performed on brain samples of European seabass fed the experimental diets. (**a**) *5ht1a* and (**b**) *drd3*. Results are expressed as mean ± SD (*n* = 5). ^a,b^ Different letters denote statistically significant differences among the experimental groups.

**Figure 5 animals-15-02060-f005:**
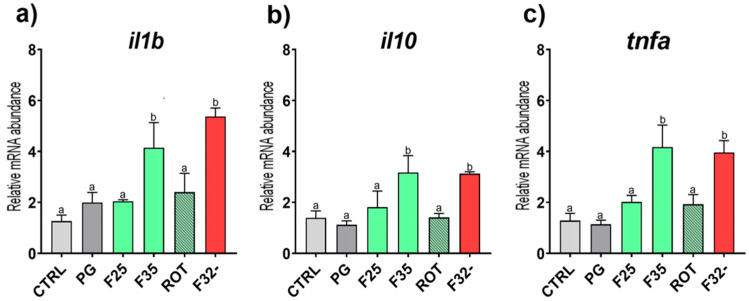
**Relative mRNA abundance of genes involved in the immune response.** Real-time qPCR was performed on distal intestinal samples of European seabass fed the experimental diets. (**a**) *il1b*, (**b**) *il10*, and (**c**) *tnfa*. Results are expressed as mean ± SD (*n* = 5). ^a,b^ Different letters denote statistically significant differences among the experimental groups.

**Table 1 animals-15-02060-t001:** Proximate composition (g/100 g) of the control diet used in the present study.

CTRL Diet	
**Proximate composition (%)**	
Crude protein %	48.00
Crude fat %	24.00
Crude fiber %	0.85
Crude ash %	8.60
Total carbohydrates %	9.55
P %	1.30
Vit C mg/kg	400
Vit E mg/kg	300

**Table 2 animals-15-02060-t002:** Score assignment criteria for mucosal fold fusion, basal inflammatory influx, and supranuclear vacuolization of enterocytes in the distal intestine.

Parameter	Score	Description
	+	0–5 observations per section
Mucosal fold fusion	++	5–15 observations per section
	+++	>15 observations per section
	+	Scarce lymphocyte infiltration
Basal inflammatory influx	++	Moderated infiltration
	+++	Diffused infiltration
	−	Absent
Supranuclear vacuoles	+	Scattered
	++	Diffused
	+++	Highly abundant

**Table 3 animals-15-02060-t003:** Sequences, annealing temperature (A.T.), and NCBI IDs of the primers used in the present study.

Gene	Forward Primer (5′–3′)	Reverse Primer (5′–3′)	A.T. (°C)	NCBI ID
*npy*	AACTCCAACAGCGCAGTACA	CGTGGGTTGTTGGTATGAGA	59	XM_051391215.1
*5ht1a*	GGGTTGTTTTCAGGACCAAG	CGCAGAAAGGAGAAGAGCAA	58	XM_051426243
*drd3*	GAATGAGCTGCGAGGTGAA	CGTGGGTTGTTGGTATGAGA	58	XM_051411236
*il1b*	AACTCCAACAGCGCAGTACA	AGACTGGCTTTGTCCACCAC	58	AJ_311925
*il10*	GCAGTCCCATGTGCAACAAC	TGCTACTGAACCTACGTCGC	59	AM_268529
*tnfa*	GACTGGCGAACAACCAGATT	GTCCGCTTCTGTAGCTGTCC	59	DQ_070246
*b-actin* (hk)	GGTACCCATCTCCTGCTCCAA	GACGTCGCACTTCATGATGCT	60	AJ_537421
*18s* (hk)	AGGGTGTTGGCAGACGTTAC	CTTCTGCCTGTTGAGGAACC	60	XM_051390998

**Table 4 animals-15-02060-t004:** Survival rate and zootechnical performance of European seabass fed the experimental diets.

	CTRL	PG	F25	F35	ROT	F32-	*p*-Value
SR (%)	100	100	100	100	100	100	-
IBW (g/fish)	67.8 ± 6.3	71.5 ± 8.9	72.6 ± 8.1	75.3 ± 7.8	71.6 ± 7.8	75.9 ± 8.3	0.8291
FBW (g/fish)	196.0 ± 35.4 ^a^	180.1 ± 17.6 ^a^	194.1 ± 15.1 ^a^	256.1 ± 16.1 ^b^	219.9 ± 19.7 ^ab^	176.2 ± 10.8 ^a^	0.0043
WG (g/fish)	120.9 ± 25.8 ^a^	108.1 ± 17.6 ^a^	122.2 ± 15.2 ^a^	194.1 ± 26.1 ^b^	167.9 ± 32.7 ^ab^	110.4 ± 13.9 ^a^	0.0026
RGR (%)	181.6 ± 22.7 ^a^	160.1 ± 24.5 ^a^	169.7 ± 21.0 ^a^	251.9 ± 20.2 ^b^	210.3 ± 27.2 ^ab^	176.6 ± 19.3 ^a^	0.0034
SGR (%)	1.7 ± 0.1 ^ab^	1.5 ± 0.2 ^ab^	1.6 ± 0.1 ^ab^	2.2 ± 0.2 ^c^	1.8 ± 0.2 ^bc^	1.3 ± 0.1 ^a^	0.0003
FCR	0.96 ± 0.08	1.17 ± 0.09	1.17 ± 0.09	0.96 ± 0.10	1.04 ± 0.09	1.20 ± 0.15	0.0359
Daily FI (%)	75.4 ± 2.7 ^a^	74.1 ± 2.2 ^a^	80.4 ± 3.2 ^ab^	90.1 ± 5.6 ^b^	87.6 ± 2.5 ^b^	73.9 ± 6.1 ^a^	0.0008

Values are shown as mean ± SD (*n* = 3). Abbreviations: SR, survival rate; IBW, initial weight; FBW, final weight; WG, weight gain; RGR, relative growth rate; SGR, specific growth rate; FCR, feed conversion ratio; FI, feed intake. ^a,b^ Different letters denote statistically significant differences among the experimental groups.

**Table 5 animals-15-02060-t005:** Histological parameters considered in distal intestine and liver of European seabass fed experimental diets.

	CTRL	PG	F25	F35	ROT	F32-	*p*-Value
Distal intestine							
Mucosal fold height	1023.0 ± 73.9	969.4 ± 52.0	977.7 ± 71.9	942.2 ± 75.2	1097.0 ± 40.2	918.0 ± 30.1	<0.05
Submucosa width	38.98 ± 0.04	40.92 ± 2.50	45.03 ± 2.64	40.31 ± 2.05	42.34 ± 1.54	45.86 ± 2.04	<0.05
Mucosal fold fusion	+	+	+	+	+	+	−
Basal inflammatory influx	+	+	+	+	+	+	−
Supranuclear vacuoles	−	−	−	−	−	−	−
Hepatic parenchyma							
Fat fraction (%)	48.9 ± 3.5 ^a^	50.7 ± 4.5 ^a^	49.7 ± 6.5 ^a^	60.5 ± 3.9 ^b^	58.5 ± 2.6 ^b^	47.8 ± 7.3 ^a^	<0.0001

Values are presented as mean ± SD (*n* = 30). ^a,b^ Different letters denote statistically significant differences among the experimental groups. + means 0–5 observations per section; − means absent.

## Data Availability

The data presented in this study are available on request from the corresponding author.

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
