# Peer review of "Synthetic Feed Attractants in European Seabass (Dicentrarchus labrax) Culture: Effects on Growth, Health, and Appetite Stimulation"

_animals, 2025, doi:10.3390/ani15142060_

Round 1

Reviewer 1 Report

Comments and Suggestions for Authors

The paper covers an important research topic, however the results are dependent upon the dietary formulation used (commercial feed formulation - not given) and the potential effect of the attractants on the final taste of the fish produced (if any) - not considered!. Moreover, 1% w/w addition is a very high amount and no indication given as to potential cost? The authors should at least say how much fishmeal and fish oil was added to the diet. Under commercial conditions diets will be extruded at high temperatures, and this aspect is not discussed. 

Author Response

Reviewer 1

The paper covers an important research topic, however the results are dependent upon the dietary formulation used (commercial feed formulation - not given) and the potential effect of the attractants on the final taste of the fish produced (if any) - not considered!. Moreover, 1% w/w addition is a very high amount and no indication given as to potential cost? The authors should at least say how much fishmeal and fish oil was added to the diet. Under commercial conditions diets will be extruded at high temperatures, and this aspect is not discussed.

We thank the reviewer for the insightful comments.

A statement in the revised manuscript, acknowledging the need for future studies, to evaluate the potential effects of flavor on flesh taste and consumer acceptance, has now been added.

Regarding feed processing, the attractants were added after the pelletizing process to avoid degradation due to high temperatures during extrusion. This method helps to preserve the bioactivity and palatability of the flavor compounds. However, we agree that further research is needed to assess the practical feasibility and stability of these attractants under commercial production conditions.

Finally, all the details about the diet’s composition have been added in the Table 1 of the revised version of the manuscript.

We trust that these revisions address the reviewer’s concerns and strengthen the methodological transparency of our study.

Reviewer 2 Report

Comments and Suggestions for Authors

The paper (Animals-3721543) under the title " Synthetic feed attractants in European seabass (Dicentrarchus labrax) culture: effects on growth, welfare, and appetite stimulation" mainly explored the detailed influence of three diets containing different synthetic flavors (two attractive, one repulsive) on growth performance, welfare, and appetite stimulation in European seabass. Diets containing three synthetic flavors had no negative influence in welfare in this study. Moreover, the diet containing 1% caramel flavor (attractive) showed superior performance in European seabass, resulting in the higher feed intake and the improved feeding response.

The main content of this paper is important to provide new insights into synthetic flavors as feed attractants in aquaculture and could provide the reference for developing effective and sustainable strategies to improve feeding and management in aquaculture.

However, several parts of this manuscript (particularly the introduction) are written in an over-long way. Also, there are some mistakes in language, syntax, and format.

Thus, the whole manuscript should be revised and proofread by a professional or native speaker.

Major comments:

1. In the "Abstract" part, key information is missing, such as the initial size and total number of European seabass. For example, Line 35. It should be clearly written and added more necessary information. Additionally, it is suggested to compress or simplify the content of "Simple Summary" part.

The current "Keywords" (Line 47) might not a good match to the main content of this manuscript. Please revise it by adding the correct terms and removing the redundant phrases. For example, "synthetic flavors" and "Dicentrarchus labrax" should be included.  

2. The current "1.Introduction" part is written in an over-long way and contains multiple lengthy and unnecessary descriptions.

For example,  the texts in the 1st (Line 49-81) and 2nd paragraph (Line 82-104) contained many unnecessary and redundant statements on research background, which may reduce the readability of the manuscript or make readers confused about the main content of this study. It is strongly suggested to compress or delete the non-core descriptions, such as Line 49-66, Line 69-80, and Line 92-96. Similar problem is present in the text of Line 115-120, in which the descriptions on zebrafish could be removed or streamlined.

Thus, the authors should simplify the relevant statement of "1.Introduction" for a clear focus on background or significance in this study.

3.In the "2.Materials and Methods" part, some descriptions on key information are incomplete and unclear.

According to the text in Line 176-177, there was a 2-week acclimation period. So, what feed were used during the adaptation period? Control feed, commercial feed or other feed? What about the aquatic environmental parameters (such as salinity, pH, dissolved oxygen, ammonia nitrogen, etc,) during the feeding experiment? These parameters were same as those in the adaptation period or not?

Thus, the author should re-check the "2.Materials and Methods" part and provide more relevant information.

4. In the "3.Results" part, the information on some tables (Table 2-3) is incomplete. For example, the meaning of superscript letters should be clarified in the legend of Table 2. Similar mistakes are present in the other table legend of the result part (Table 3).

Additionally, the original texts in Line 272-273 contained unnecessary statements on the results. They could be compressed or deleted directly without any negative influence on the corresponding paragraph.  

5. The current version is a bit chaotic, including wrong/missing volume and page numbers.

For example, in Reference 5, 19, 26, 28, etc, page number is missing. Reference 47 and Reference 58 19 have the wrong information on page number. Similarly, the information on volume number should be included in Reference 25, 42, etc.

Additionally, the cited literatures published in 2020-2025 are less than 24 (total literatures: 61). Please make sure about 50% of the references are within 5 years (2020-2025).

Thus, the authors need to re-check the reference list according to the instructions for authors and revise accordingly.

Other errors (highlighted in yellow) were marked in the PDF file.

So, this manuscript will be reconsidered after minor revision.

Author Response

Reviewer 2

The paper (Animals-3721543) under the title " Synthetic feed attractants in European seabass (Dicentrarchus labrax) culture: effects on growth, welfare, and appetite stimulation" mainly explored the detailed influence of three diets containing different synthetic flavors (two attractive, one repulsive) on growth performance, welfare, and appetite stimulation in European seabass. Diets containing three synthetic flavors had no negative influence in welfare in this study. Moreover, the diet containing 1% caramel flavor (attractive) showed superior performance in European seabass, resulting in the higher feed intake and the improved feeding response.

The main content of this paper is important to provide new insights into synthetic flavors as feed attractants in aquaculture and could provide the reference for developing effective and sustainable strategies to improve feeding and management in aquaculture.

However, several parts of this manuscript (particularly the introduction) are written in an over-long way. Also, there are some mistakes in language, syntax, and format.

Thus, the whole manuscript should be revised and proofread by a professional or native speaker.

The manuscript has been reviewed for English.

Major comments:

  1. In the "Abstract" part, key information is missing, such as the initial size and total number of European seabass. For example, Line 35. It should be clearly written and added more necessary information. Additionally, it is suggested to compress or simplify the content of "Simple Summary" part.

The current "Keywords" (Line 47) might not a good match to the main content of this manuscript. Please revise it by adding the correct terms and removing the redundant phrases. For example, "synthetic flavors" and "Dicentrarchus labrax" should be included. 

 We thank the reviewer for these helpful suggestions. We have revised the Abstract section to include the initial weight, and the number of fish used in the study. The Simple Summary section has been shortened and simplified for clarity. The terms “synthetic flavors” and “Dicentrarchus labrax” are present in the Title and, as known, should not be included in the keywords. We hope these revisions address the reviewer’s concerns and improve the manuscript's readability and scientific transparency.

  1. The current "1.Introduction" part is written in an over-long way and contains multiple lengthy and unnecessary descriptions.

For example,  the texts in the 1st (Line 49-81) and 2nd paragraph (Line 82-104) contained many unnecessary and redundant statements on research background, which may reduce the readability of the manuscript or make readers confused about the main content of this study. It is strongly suggested to compress or delete the non-core descriptions, such as Line 49-66, Line 69-80, and Line 92-96. Similar problem is present in the text of Line 115-120, in which the descriptions on zebrafish could be removed or streamlined.

Thus, the authors should simplify the relevant statement of "1.Introduction" for a clear focus on background or significance in this study.

 We appreciate the reviewer’s insightful comments. The Introduction section has been thoroughly revised to improve clarity and focus, and non-essential background information were removed or condensed in the specified lines. However, the zebrafish discussion (Line 115-120, which have now become 124-129 in the revised version of the Manuscript) has been maintained for the relevance of the current study. In fact, the references to zebrafish studies (Conti et al., 2023-2024) were included since they specifically relate to previous works testing the same flavors used in the current study, providing basis for comparison. We hope these changes and clarification aim to enhance readability and better highlight the study's significance.

3.In the "2.Materials and Methods" part, some descriptions on key information are incomplete and unclear.

According to the text in Line 176-177, there was a 2-week acclimation period. So, what feed were used during the adaptation period? Control feed, commercial feed or other feed? What about the aquatic environmental parameters (such as salinity, pH, dissolved oxygen, ammonia nitrogen, etc,) during the feeding experiment? These parameters were same as those in the adaptation period or not?

Thus, the author should re-check the "2.Materials and Methods" part and provide more relevant information.

We thank the reviewer for these observations. During the 2-week acclimation period, fish were fed the control (CTRL) diet. The aquatic environmental parameters were maintained consistently and remained the same throughout both the acclimation and experimental feeding periods. We have clarified this point in the revised version of the manuscript, specifically in the “2.3 Experimental design and zootechnical parameters” section.

  1. In the "3.Results" part, the information on some tables (Table 2-3) is incomplete. For example, the meaning of superscript letters should be clarified in the legend of Table 2. Similar mistakes are present in the other table legend of the result part (Table 3).

Additionally, the original texts in Line 272-273 contained unnecessary statements on the results. They could be compressed or deleted directly without any negative influence on the corresponding paragraph. 

We thank the reviewer for highlighting these points. We have revised the legends of Table 2 and Table 3 to clearly explain the meaning of the superscript letters used for statistical comparisons. Additionally, we have reviewed the text of Lines 272-273 and compressed the statements as suggested, to improve clarity and conciseness.

  1. The current version is a bit chaotic, including wrong/missing volume and page numbers.

For example, in Reference 5, 19, 26, 28, etc, page number is missing. Reference 47 and Reference 58 19 have the wrong information on page number. Similarly, the information on volume number should be included in Reference 25, 42, etc.

Additionally, the cited literatures published in 2020-2025 are less than 24 (total literatures: 61). Please make sure about 50% of the references are within 5 years (2020-2025).

Thus, the authors need to re-check the reference list according to the instructions for authors and revise accordingly.

Thank the reviewer for his careful review of the reference list. We have verified the citations and confirm that some references without page numbers are correct as per the source format (e.g., articles published online or book chapters). Nevertheless, we will carefully check and ensure that all references comply with the journal’s formatting requirements before final submission. We have also reviewed the reference list and incorporated several recent studies to strengthen the manuscript’s relevance. However, older references are, in our opinion, important for this study.

Other errors (highlighted in yellow) were marked in the PDF file.

We thank the reviewer for all the suggestions.

So, this manuscript will be reconsidered after minor revision.

Reviewer 3 Report

Comments and Suggestions for Authors

Dear Authors,
Thank you for the opportunity to review your manuscript entitled “Synthetic feed attractants in European seabass (Dicentrarchus labrax) culture: effects on growth, welfare, and appetite stimulation.” Please find attached a detailed review of each section of the manuscript, including critical observations and suggestions aimed at enhancing the clarity, methodological consistency, and scientific robustness of the study. I hope the comments provided will contribute constructively to the improvement of your research.

Title

  • The use of the term “welfare” is not clearly defined in the manuscript: classical welfare indicators (such as behaviour, stress, low mortality, etc.) are not measured, except for histological evaluation. It might be more appropriate to replace it with “health” or to specify what is understood by “welfare” in this context.

Abstract

  • Numerical details are missing from the results. For instance, it is mentioned that F35 significantly increased feed intake and growth, but percentages or concrete comparative values should be provided.

  • Including key elements of the experimental design in the abstract (e.g., number of fish, tank replicates) would be helpful. Omitting these figures hinders the reader’s ability to assess the study's robustness at a glance.

  • Adding a clear opening sentence on the rationale (e.g., the need for synthetic attractants) would improve understanding from the outset.

Introduction

  • The introduction is somewhat lengthy. The information on alternative raw materials and economic aspects could be condensed to focus more directly on the topic of feed attractants.

  • Challenges related to palatability and the general role of attractants are clearly identified. However, relevant previous studies on seabass that support the rationale are missing.

  • The link between the omnivorous model species (zebrafish) and the carnivorous target species is based on the assumption of shared sensory mechanisms. It would be helpful to briefly explain the physiological basis for this assumption or cite comparative evidence (e.g., studies comparing olfactory responses or feeding behaviour in species with different diets).

  • The hypothesis or research questions are not clearly defined at the end of the introduction. While the general aim is mentioned, it could be more explicitly formulated.

  • Furthermore, it is important to clarify how “welfare” will be interpreted within the framework of the study, in order to avoid ambiguity.

Materials and Methods

  • Ethics: Compliance with European regulations and approval by an ethics committee is correctly stated and well documented.

  • Diet preparation: The preparation of synthetic flavours and their addition (1% in propylene glycol) is described. However, the basic nutritional composition (protein, fat, carbohydrates) of the commercial base feed should be included directly rather than referred to via a web link. This would facilitate reproducibility and interpretation. The use of a micropipette for flavour application is acceptable, but it would be advisable to explain how homogeneity was ensured (e.g., volume applied per batch, drying of feed after application).

  • Experimental design: Six groups were used (CTRL, PG, F25, F35, F32–, and ROT), each with three tank replicates. The random assignment of fish to tanks is stated, which is appropriate. However, the fixed feeding regime at 1.5% of body weight without ad libitum feeding limits the capacity to measure voluntary feed intake. Although justified based on commercial practices, it would be preferable to consider ad libitum feeding with leftover collection to assess maximum feeding motivation. The choice of 15 minutes for feed recovery is based on a previous test; this should be emphasised as a methodological decision.

  • Histological analysis: Sampling of 10 fish per tank (n = 30 per group) is adequate. Fixation in Bouin’s solution and routine staining (H&E, PAS) are standard. However, it is not stated how sampling was performed (e.g., simple randomisation? blinded to treatment?) to avoid bias. Also, quantitative or semi-quantitative criteria for histopathological scoring are not described, beyond citing Zarantoniello et al. [48]; these criteria should be summarised in a table or supplement for transparency.

  • Molecular analysis: Key genes (npy, drd3, 5ht1a, il1b, il10, tnfa) were analysed using β-actin and 18S as reference genes via qPCR. It should be clarified which reference gene(s) were used for final normalisation (both? geometric mean?). According to MIQE guidelines, reference gene stability should be assessed and at least two validated genes used; if so, this must be stated. Additionally, Table 1 contains an error: 5ht1a is listed twice with the same sequence, indicating an editing mistake (possibly a missing gene), raising concerns about data integrity. This should be corrected. It would also be helpful to specify which brain region was sampled (e.g., telencephalon, diencephalon), given that appetite regulation in fish is centred in the hypothalamus.

  • Statistical analysis: One-way ANOVA with Tukey’s post hoc test (p < 0.05) was used. However, a conceptual problem arises: the experimental unit is the tank, not the individual fish. The statistical unit and whether ANOVA degrees of freedom correspond to tanks (n = 3) or fish (n = 30) should be clearly stated.

Results

  • Zootechnical performance: The presentation (Table 2) is clear, but there is an inconsistency: the text states that no significant differences in FCR were found among groups, while the table reports a p-value of 0.0359. This contradiction must be clarified (perhaps a mislabelled significance letter?). The authors should verify the statistical analysis of FCR.

  • A 100% survival rate is reported for all groups, which is unusual for a 90-day trial; it would be advisable to indicate how any mortality was handled (was there truly no loss?).

  • Histological analysis: It is stated that no significant morphological differences were found in the distal intestine, but p-values are not provided, nor are means with standard deviations reported in the text (only in Table 3, with dashes under “p-value”). P-values should be reported or the omission should be justified.

  • Conversely, a notable effect was observed in the liver: F35 and ROT groups had significantly higher fat fractions. Although this is interpreted as an energetic response, such lipid accumulation is a relevant finding that, despite no signs of inflammation, indicates a physiological response to increased caloric intake. This should be reported in the Results section, not only discussed later.

  • In the liver photomicrographs with PAS staining, only one image is annotated with "asterisk indicates glycogen deposition”. However, other important histological components present should also be indicated.

  • Gene expression: The immune-related gene upregulation (il1b, il10, tnfa) in F35 and F32- groups is stated, with no differences between them. It would be helpful to quantify the magnitude of upregulation. The ROT group also showed intermediate or divergent responses compared to F25/F35; this deserves at least brief mention in the Results, even if elaborated upon in the Discussion.

  • Finally, sample size (e.g., n = 5) should always be specified, and graphical representations should clearly indicate which treatments differ statistically (letter annotations).

Discussion

  • The discussion effectively summarises findings and relates them to the literature, though at times it becomes lengthy and speculative. It correctly highlights that F35 (caramel) increased feed intake and growth, consistent with results in zebrafish. However, the absence of negative effects (labelled as “welfare”) should be interpreted with greater caution given the marked hepatic fat accumulation. Although no visible inflammation was observed, lipid accumulation could pose long-term health risks or reflect excessive energy intake; this warrants discussion (e.g., potential effects on future reproductive performance or stress resistance).

  • The upregulation of proinflammatory genes (il1β, tnfa) in F35 and F32- is attributed to extraplasmic functions of cytokines. While this is a valid interpretation, it should be complemented by considering that a subtle immune response, not visible in histology, may have occurred, or that the solvent (propylene glycol) alone was not responsible (as the PG group was comparable to CTRL). The magnitude of these gene changes and their potential relation to flavour compound absorption, alongside the absence of morphological alterations, should be discussed.

  • The reduced expression of npy is appropriately linked to increased satiety. To strengthen this, additional literature on npy/5ht postprandial trends in fish (e.g., Lubzens et al.) could be cited. Similarly, the interpretation of the monoaminergic system (5-HT1A and DRD3) is interesting but should be approached with caution: for instance, it is assumed that lower drd3 in F35 indicates hedonic adaptation. While plausible, supporting evidence from previous fish studies should be cited.

  • The section on the ROT group and the effect of the “cheese” flavour (F25) is somewhat vague. Given F25’s lower effectiveness, it should be considered whether the base compound (F25) is genuinely less attractive to seabass, or whether the rotation scheme diluted the stimulus. A possible improvement would be to suggest future preference tests (e.g., multiple-choice feeding trials) to validate flavour preferences.

  • It would be appropriate to acknowledge practical limitations, such as the cost of synthetic flavour production or their possible absorption/dispersal in water, which were not addressed.

  • Overall, the tone of the conclusions should be moderated: although F35 clearly improved zootechnical performance, the term “welfare” used in the title is not comprehensively supported by the data. It would be more accurate to state that no significant histological adverse effects were observed (within the measured parameters), rather than asserting global welfare.

References

  • Most are up-to-date and relevant. However, key references in English or Spanish regarding similar cases in D. labrax are missing. In addition to Dias et al. (1997), recent studies on anglerfish or other carnivorous species using synthetic or volatile attractants could be cited.

  • The authors support their rationale with recent references (e.g., Conti et al., Zebrafish 2022–2023), which is commendable. Nevertheless, minor formatting issues should be corrected (e.g., duplication of the 5ht1a gene in the primer table), and all in-text citations ([38], [45], etc.) should match the reference list.

Overall Assessment
The manuscript addresses a relevant topic in modern aquaculture. The findings indicate that the addition of caramel flavour (F35) increased feed intake and growth in seabass without causing evident histopathological alterations, which is valuable. However, there are significant methodological and presentational shortcomings, including errors in the primer table, contradictions in reported results (e.g., FCR), and omission of key prior studies. Additionally, the interpretation of “welfare” is overly broad given the lack of specific indicators. These aspects require correction. With the proposed improvements, the study could contribute meaningfully to the literature on fish nutrition and feeding behaviour.

Kind regards,

Author Response

Reviewer 3

Dear Authors,

Thank you for the opportunity to review your manuscript entitled “Synthetic feed attractants in European seabass (Dicentrarchus labrax) culture: effects on growth, welfare, and appetite stimulation.” Please find attached a detailed review of each section of the manuscript, including critical observations and suggestions aimed at enhancing the clarity, methodological consistency, and scientific robustness of the study. I hope the comments provided will contribute constructively to the improvement of your research.

Title

The use of the term “welfare” is not clearly defined in the manuscript: classical welfare indicators (such as behaviour, stress, low mortality, etc.) are not measured, except for histological evaluation. It might be more appropriate to replace it with “health” or to specify what is understood by “welfare” in this context.

We appreciate this insightful comment. We agree that the term "welfare" as used in the manuscript could benefit from clarification. Since classical welfare indicators such as behaviour and physiological stress markers were not assessed, we have revised the term “welfare” where appropriate and replaced it with “health” to better reflect the scope of our findings.

Abstract

Numerical details are missing from the results. For instance, it is mentioned that F35 significantly increased feed intake and growth, but percentages or concrete comparative values should be provided.

Including key elements of the experimental design in the abstract (e.g., number of fish, tank replicates) would be helpful. Omitting these figures hinders the reader’s ability to assess the study's robustness at a glance.

Adding a clear opening sentence on the rationale (e.g., the need for synthetic attractants) would improve understanding from the outset.

We thank the reviewer for these constructive suggestions. We have revised the abstract and results section to include specific numerical values for feed intake and growth improvements with F35. Key experimental design details (number of fish, replicates, trial duration) have also been added to the abstract. Additionally, we introduced a clear opening sentence to highlight the rationale for using synthetic attractants. We hope these revisions address the reviewer’s concerns and improve the manuscript's readability and scientific transparency.

Introduction

The introduction is somewhat lengthy. The information on alternative raw materials and economic aspects could be condensed to focus more directly on the topic of feed attractants.

We thank the reviewer for this observation. In response, we have revised and shortened the paragraphs. The revised Introduction section now focuses more directly on the main objectives of the study.

Challenges related to palatability and the general role of attractants are clearly identified. However, relevant previous studies on seabass that support the rationale are missing.

We appreciate the reviewer’s valuable comment. To strengthen the rationale of our study, we have now included references to previous studies conducted on seabass in the Introduction section (e.g., Lanari et al., 2005; Torrecillas et al., 2017; Resende et al., 2024).

The link between the omnivorous model species (zebrafish) and the carnivorous target species is based on the assumption of shared sensory mechanisms. It would be helpful to briefly explain the physiological basis for this assumption or cite comparative evidence (e.g., studies comparing olfactory responses or feeding behaviour in species with different diets).

We thank the reviewer for this constructive observation. To clarify the rationale for using zebrafish as a preliminary model, we have added a brief explanation in the manuscript regarding the physiological similarities in olfactory and neuroendocrine mechanisms across teleost fish (specifically in Lines 131-136, in the revised version of the manuscript).

The hypothesis or research questions are not clearly defined at the end of the introduction. While the general aim is mentioned, it could be more explicitly formulated.

We thank the reviewer for the suggestion. We have revised the final paragraph of the Introduction to more clearly and explicitly state the work’s objective. The updated text now better defines the research question and highlights the rationale of the study.

Furthermore, it is important to clarify how “welfare” will be interpreted within the framework of the study, in order to avoid ambiguity.

Please see the Title comment.

Materials and Methods

Ethics: Compliance with European regulations and approval by an ethics committee is correctly stated and well documented.

Diet preparation: The preparation of synthetic flavours and their addition (1% in propylene glycol) is described. However, the basic nutritional composition (protein, fat, carbohydrates) of the commercial base feed should be included directly rather than referred to via a web link. This would facilitate reproducibility and interpretation. The use of a micropipette for flavour application is acceptable, but it would be advisable to explain how homogeneity was ensured (e.g., volume applied per batch, drying of feed after application).

We thank the reviewer for this helpful suggestion. The proximate nutritional composition of the commercial base diet has now been included directly in the manuscript, in Table 1 (in “2.2 Synthetic flavors and production of experimental diets” section). The explanation of how homogeneity was ensured for flavors incorporation in the feed was already reported in the original manuscript (Line 191).

Experimental design: Six groups were used (CTRL, PG, F25, F35, F32–, and ROT), each with three tank replicates. The random assignment of fish to tanks is stated, which is appropriate. However, the fixed feeding regime at 1.5% of body weight without ad libitum feeding limits the capacity to measure voluntary feed intake. Although justified based on commercial practices, it would be preferable to consider ad libitum feeding with leftover collection to assess maximum feeding motivation. The choice of 15 minutes for feed recovery is based on a previous test; this should be emphasised as a methodological decision.

We acknowledge the reviewer’s concern regarding the fixed feeding regime. In this study, a feeding rate of 1.5% of body weight per day was chosen to reflect commercial aquaculture practices, where controlled rations are commonly used to optimize feed conversion and minimize feed waste. Moreover, a fixed feeding eliminates inter-individual or tank-level variation in feed intake, due to dominant-subordinate dynamics or behavioral variability. While this approach limits the direct assessment of voluntary feed intake, it allows for standardized comparison of feed attractant efficacy under industry-relevant conditions. We agree that ad libitum feeding with leftover collection could provide additional insight into feeding motivation and we will consider this approach in future studies to complement our findings. The feeding approach used in the present study is further supported by Turchini and Hardy, 2024.
The choice of 15 minutes for feed recovery was already reported in the manuscript and now emphasized as a methodological decision (Line 226).

Histological analysis: Sampling of 10 fish per tank (n = 30 per group) is adequate. Fixation in Bouin’s solution and routine staining (H&E, PAS) are standard. However, it is not stated how sampling was performed (e.g., simple randomisation? blinded to treatment?) to avoid bias. Also, quantitative or semi-quantitative criteria for histopathological scoring are not described, beyond citing Zarantoniello et al. [48]; these criteria should be summarised in a table or supplement for transparency.

We thank the reviewer for this important observation. In the revised manuscript, we have clarified the sampling procedure. Specifically, fish selected for the different analyses were randomly sampled from each tank by a blinded operator to the dietary treatments to minimize sampling bias. Moreover, histopathological scoring criteria have now been added in the revised version of the manuscript, in the Table 2 (“2.4 Histological analysis” section).

Molecular analysis: Key genes (npy, drd3, 5ht1a, il1b, il10, tnfa) were analysed using β-actin and 18S as reference genes via qPCR. It should be clarified which reference gene(s) were used for final normalisation (both? geometric mean?). According to MIQE guidelines, reference gene stability should be assessed and at least two validated genes used; if so, this must be stated. Additionally, Table 1 contains an error: 5ht1a is listed twice with the same sequence, indicating an editing mistake (possibly a missing gene), raising concerns about data integrity. This should be corrected. It would also be helpful to specify which brain region was sampled (e.g., telencephalon, diencephalon), given that appetite regulation in fish is centred in the hypothalamus.

We thank the reviewer for these insightful suggestions.

In accordance with MIQE guidelines, we evaluated the stability of the two candidate reference genes (β-actin and 18s) across all experimental groups, and both genes were used for final normalization as the geometric mean of their Ct values. This information has now been included in the revised manuscript, specifically in the “2.5 Molecular analyses” section.
Additionally, the duplication of the 5ht1a gene in Table 1 was an unintentional editing error. We confirm that no additional gene was omitted from the analysis or data, and the qPCR was correctly performed on the six intended target genes: npy, drd3, 5ht1a, il1b, il10, and tnfa. The duplicated 5ht1a row has been removed in the revised version of Table 1 (now Table 3 in the revised manuscript).

Finally, in this study, the whole brain was sampled rather than isolating specific regions, to provide a global overview of central gene expression patterns. This clarification has now been reported in the revised manuscript.
We appreciate the reviewer’s attention to these methodological details, and we hope the revised version now fully addresses the concerns raised.

Statistical analysis: One-way ANOVA with Tukey’s post hoc test (p < 0.05) was used. However, a conceptual problem arises: the experimental unit is the tank, not the individual fish. The statistical unit and whether ANOVA degrees of freedom correspond to tanks (n = 3) or fish (n = 30) should be clearly stated.

We thank the reviewer for this important observation. The tanks (n = 3 per treatment) were used as the experimental unit for analyses related to survival rate and zootechnical performance. For these parameters, ANOVA and Tukey’s post hoc were performed using the tank means, with degrees of freedom corresponding to the number of tanks. For all other analyses (i.e., molecular, histological), individual fish were considered as experimental units (n = 15 or n = 30). We have clarified this point in the Methods section to avoid any confusion and ensure proper interpretation of the results (“2.6 Statistical analysis”).

Results

Zootechnical performance: The presentation (Table 2) is clear, but there is an inconsistency: the text states that no significant differences in FCR were found among groups, while the table reports a p-value of 0.0359. This contradiction must be clarified (perhaps a mislabelled significance letter?). The authors should verify the statistical analysis of FCR.

We thank the reviewer for highlighting this discrepancy. The reported ANOVA p-value for FCR (0.0359) indicates an overall significant difference among groups. However, subsequent multiple comparison tests did not identify any significant pairwise differences between groups. This can occur when the overall variability is sufficient for ANOVA significance, but individual group conversely are not strong enough to reach significance after correction for multiple testing. This pattern suggests the presence of subtle overall group differences without clear pairwise contrasts under the multiple testing correction applied. To clarify this aspect, we have revised the text in the Results section, reporting that while the overall ANOVA was significant, post-hoc comparisons did not detect significant differences in FCR between any specific groups.

A 100% survival rate is reported for all groups, which is unusual for a 90-day trial; it would be advisable to indicate how any mortality was handled (was there truly no loss?).

Throughout the 90-day trial, all experimental groups exhibited a 100% survival rate, with no mortality observed at any point.

Histological analysis: It is stated that no significant morphological differences were found in the distal intestine, but p-values are not provided, nor are means with standard deviations reported in the text (only in Table 3, with dashes under “p-value”). P-values should be reported or the omission should be justified.

We thank the reviewer for this important observation. We apologize for the oversight regarding the p-values for the distal intestine morphology. We have corrected this point in the revised version by clearly reporting the p-values and acknowledging the presence of statistically significant differences among groups, for both “mucosal folds height” and “submucosa width”.
However, we would like to clarify that although the differences were statistically significant, all values remained within the normal physiological range for Dicentrarchus labrax, as reported in the literature (e.g., Ferreira et al., 2023), thereby not reflecting pathological alterations.

Additionally, histological evaluation confirmed that the mucosal layer presented a regular columnar epithelium with polarized and basally located nuclei, with a well-preserved architecture and no visible tissue damage or inflammation. We have clarified this point in the revised manuscript and have expanded the text to better reflect both the statistical and biological interpretation of the findings.
Mucosal fold fusion, Basal inflammatory influx, and Supranuclear vacuoles were reported as a qualitative result, as reported in Table 2, and in accord to Zarantoniello et al., 2023. Thus, no p-Values can be reported.

Conversely, a notable effect was observed in the liver: F35 and ROT groups had significantly higher fat fractions. Although this is interpreted as an energetic response, such lipid accumulation is a relevant finding that, despite no signs of inflammation, indicates a physiological response to increased caloric intake. This should be reported in the Results section, not only discussed later.

We thank the reviewer for this observation. We agree that the increase in hepatic fat content is a relevant physiological finding. This information was already reported in Table 3 of the Results section (3.2), in the original manuscript (now referred as Table 5), as well as in the text at Line 349.

In the liver photomicrographs with PAS staining, only one image is annotated with "asterisk indicates glycogen deposition”. However, other important histological components present should also be indicated.

We thank the reviewer for this helpful suggestion. As the key histological components were already described and annotated in the hematoxylin and eosin Y (EE) stained sections, we chose to focus the PAS-stained image annotation specifically on glycogen deposition, which was the principal feature assessed with this staining.

Gene expression: The immune-related gene upregulation (il1b, il10, tnfa) in F35 and F32- groups is stated, with no differences between them. It would be helpful to quantify the magnitude of upregulation. The ROT group also showed intermediate or divergent responses compared to F25/F35; this deserves at least brief mention in the Results, even if elaborated upon in the Discussion.

We thank the reviewer for the comment. We respectfully note that the upregulation of immune-related genes in F35 and F32- groups, as well as the intermediate response observed in the ROT group, is already described in the manuscript. Specifically, ROT group showed no significant differences in expression levels compared to CTRL, PG, and F25. Additionally, the magnitude of gene expression changes is presented in the Results section and clearly illustrated as relative mRNA abundance in the corresponding figures, allowing for direct comparison among all experimental groups.

Finally, sample size (e.g., n = 5) should always be specified, and graphical representations should clearly indicate which treatments differ statistically (letter annotations).

We appreciate the reviewer’s observation. In the revised version of the manuscript, all figure and table captions have been updated to clarify that different superscript letters denote statistically significant differences among groups (p < 0.05). Furthermore, the sample size (n) for each analysis is now clearly specified to enhance transparency and ensure reproducibility.

Discussion

The discussion effectively summarises findings and relates them to the literature, though at times it becomes lengthy and speculative. It correctly highlights that F35 (caramel) increased feed intake and growth, consistent with results in zebrafish. However, the absence of negative effects (labelled as “welfare”) should be interpreted with greater caution given the marked hepatic fat accumulation. Although no visible inflammation was observed, lipid accumulation could pose long-term health risks or reflect excessive energy intake; this warrants discussion (e.g., potential effects on future reproductive performance or stress resistance).

We thank the Reviewer for this valuable observation. In the revised version of the manuscript, we have carefully addressed this point by including a more cautious and balanced interpretation of the increased hepatic fat accumulation observed in the F35 and ROT groups. This addition ensures a more comprehensive interpretation of the findings in the context of fish welfare (Lines: 446-451).

The upregulation of proinflammatory genes (il1β, tnfa) in F35 and F32- is attributed to extraplasmic functions of cytokines. While this is a valid interpretation, it should be complemented by considering that a subtle immune response, not visible in histology, may have occurred, or that the solvent (propylene glycol) alone was not responsible (as the PG group was comparable to CTRL). The magnitude of these gene changes and their potential relation to flavour compound absorption, alongside the absence of morphological alterations, should be discussed.

We thank the reviewer for this valuable comment. Although upregulation of proinflammatory genes was observed in the F35 and F32- groups without corresponding histological changes, subtle immune modulation at the molecular level may still occur. Artificial dietary additives, including synthetic flavors, may influence immune responses in fish by modulating gut microbiota, leading to variable outcomes as reported in the literature (e.g., García Beltrán et al., 2022). However, due to limited data, it remains difficult to generalize beneficial or negative effects of synthetic flavors on fish gut health. Given the absence of morphological changes and the moderate magnitude of gene upregulation in our study, these responses likely reflect a physiological adaptation rather than pathology. A corresponding discussion highlighting these points has been included in the revised manuscript, specifically in the Discussion section (Lines: 422-428).

The reduced expression of npy is appropriately linked to increased satiety. To strengthen this, additional literature on npy/5ht postprandial trends in fish (e.g., Lubzens et al.) could be cited. Similarly, the interpretation of the monoaminergic system (5-HT1A and DRD3) is interesting but should be approached with caution: for instance, it is assumed that lower drd3 in F35 indicates hedonic adaptation. While plausible, supporting evidence from previous fish studies should be cited.

We thank the reviewer for the insightful comment. We have now revised the manuscript to better clarify the interpretation of the downregulation of drd3 in the caramel (F35) group (Lines: 480-482). Relevant literature supporting this interpretation has also been added to strengthen the discussion, despite evidence for this mechanism in fish is limited.

Unfortunately, we were not able to find the paper by Lubzens et al.

The section on the ROT group and the effect of the “cheese” flavour (F25) is somewhat vague. Given F25’s lower effectiveness, it should be considered whether the base compound (F25) is genuinely less attractive to seabass, or whether the rotation scheme diluted the stimulus. A possible improvement would be to suggest future preference tests (e.g., multiple-choice feeding trials) to validate flavour preferences.

Thank the reviewer for his valuable suggestion. We have added a statement in the Discussion recommending future preference tests to better elucidate seabass flavor preferences and validate our findings (Lines: 494-496).

It would be appropriate to acknowledge practical limitations, such as the cost of synthetic flavour production or their possible absorption/dispersal in water, which were not addressed.

Thank the reviewer for the interesting comment. Unfortunately, these aspects were not addressed experimentally in this study. Please see Lines 114–123, in which the broader cost-effectiveness of synthetic flavors in aquaculture is addressed, and in the Conclusions section.

Overall, the tone of the conclusions should be moderated: although F35 clearly improved zootechnical performance, the term “welfare” used in the title is not comprehensively supported by the data. It would be more accurate to state that no significant histological adverse effects were observed (within the measured parameters), rather than asserting global welfare.

We thank the reviewer for this important observation. We have revised the conclusions to moderate the use of the term “welfare,” clarifying that no significant histological adverse effects were observed in key metabolic organs such as the liver and intestine. The revised text now better reflects these nuances.

References

Most are up-to-date and relevant. However, key references in English or Spanish regarding similar cases in D. labrax are missing. In addition to Dias et al. (1997), recent studies on anglerfish or other carnivorous species using synthetic or volatile attractants could be cited.

The authors support their rationale with recent references (e.g., Conti et al., Zebrafish 2022–2023), which is commendable. Nevertheless, minor formatting issues should be corrected (e.g., duplication of the 5ht1a gene in the primer table), and all in-text citations ([38], [45], etc.) should match the reference list.

Overall Assessment

The manuscript addresses a relevant topic in modern aquaculture. The findings indicate that the addition of caramel flavour (F35) increased feed intake and growth in seabass without causing evident histopathological alterations, which is valuable. However, there are significant methodological and presentational shortcomings, including errors in the primer table, contradictions in reported results (e.g., FCR), and omission of key prior studies. Additionally, the interpretation of “welfare” is overly broad given the lack of specific indicators. These aspects require correction. With the proposed improvements, the study could contribute meaningfully to the literature on fish nutrition and feeding behaviour.

Kind regards,

Reviewer 4 Report

Comments and Suggestions for Authors

Dear authors,

I send you my suggestions.

Author Response

Reviewer 4

Synthetic feed attractants in European seabass (Dicentrarchus labrax) culture: effects on growth, welfare, and appetite stimulation

Line 154: The authors mention Conti et al. [38] when discussing the composition of synthetic attractants. It would be advisable to include this description in the manuscript itself or, alternatively, as supplementary material. For the methodology to be replicable, it is essential that it contains the minimum information required in the methodology.

We thank the reviewer for this suggestion. We have included a concise description of the composition of the synthetic attractants within the Materials and Methods section (Lines 169–174).

Line 166-177: Was the commercial diet used the extruded type? Was any previous centesimal analysis carried out to determine protein and energy levels? Also, after adding the flavorings, what procedure was adopted to ensure the homogenization and uniformity of the diet? Considering that alcohol was used as the vehicle and that the flavorings were liquid blends, was any waiting time adopted for drying? Also, how was the methodology for adding the flavorings to the diets defined? Based on what evidence and tests did the authors define the 1% value for incorporating flavorings? Are there any other studies on species used in commercial aquaculture?

We thank the reviewer for these thoughtful and constructive observations. Yes, the commercial diet used in the trial was an extruded type. Centesimal composition is now reported in Table 1.

The liquid flavorings were added post-pelletization and homogenized by manual mixing ensuring uniform distribution, as already performed in previous studies (Conti et al., 2023-2024). A resting time of 24 hours at room temperature was adopted to ensure proper absorption. The incorporation rate of 1% w/w was selected based on previous studies and aligned with concentrations used for synthetic flavors in fish studies (Conti et al., 2023-2024; Lucon-Xiccato et al., 2020). These references have now been cited in the revised manuscript and a detailed description has been added in the revised version of the manuscript (in “2.2 Synthetic flavors and production of experimental diets” section).

Line 183-186: What criteria or evidence was used to define the weekly rotation of attractive diets (F25 and F35)? Was this interval considered sufficient to avoid possible adaptation of the olfactory receptors?

The weekly rotation scheme was chosen based on previous studies suggesting that olfactory receptor adaptation or habituation to odorant stimuli can occur within days to weeks (e.g., Barry et al., 2003; Miller-Sims et al., 2011). A one-week interval was therefore considered sufficient to minimize sensory adaptation and maintain the effectiveness of the attractants throughout the trial. The above-mentioned references have now been added to the revised version of the manuscript to improve the clarity of the study.

Line 189: For the study with flavorings, how did the authors come to the conclusion that the best way to provide the diet would be by the percentage of biomass and not by apparent satiety?

We acknowledge the reviewer’s concern regarding the fixed feeding regime. In this study, a feeding rate of 1.5% of body weight per day was chosen to reflect commercial aquaculture practices, where controlled rations are commonly used to optimize feed conversion and minimize waste. Moreover, a fixed feeding eliminates inter-individual or tank-level variation in feed intake, due to dominant-subordinate dynamics or behavioral variability. While this approach limits the direct assessment of voluntary feed intake, it allows for standardized comparison of feed attractant efficacy under industry-relevant conditions. We agree that ad libitum feeding with leftover collection could provide additional insight into feeding motivation and we will consider this approach in future studies to complement our findings. The feeding approach used in the present study is further supported by Turchini and Hardy, 2024.

Line 255-258: It's not clear which calculation was used for gene expression?

Line 289-292, in the revised version of the manuscript: “The qPCR data were processed using the iQ5 optical system software version 2.0 (Bio-Rad), including GeneEx Macro iQ5 Conversion and genex Macro iQ5 files. Gene transcript expression variations among experimental groups are reported as relative mRNA abundance (arbitrary units).”

Line 388-389: This paper makes several references to studies with Zebrafish. However, there have been relevant publications in the last five years involving species of interest in aquaculture and the use of dietary attractants. Why were these studies not cited?

We thank the reviewer for this insightful comment. The references to zebrafish studies (Conti et al., 2023–2024) were included specifically because these previous experiments tested the same flavors used in the present study, providing a fundamental basis before application to a commercially relevant species such as seabass. While we acknowledge the existence of recent relevant studies on dietary attractants in aquaculture species, our focus in that specific part of the Discussion section was deliberately built upon the prior zebrafish research as a foundational model.
However, throughout the text, a number of papers related to aquaculture species have been included.

Line 392-395: How can the accumulation of fat observed in the liver of the group given the ROT diet be explained, considering that the diets were isolipidic? What metabolic pathways are possibly involved in this mechanism? Why weren't the viscerosomatic and hepatosomatic indices done?

We appreciate the reviewer’s comment. Although the diets were isolipidic, the increased hepatic fat accumulation observed in the ROT group is likely a consequence of increased feed intake, leading to higher energy accumulation and lipid deposition. Viscerosomatic and hepatosomatic indices were not assessed but are now acknowledged as important parameters for future studies.

Line 399-401: Higher growth is not always associated with higher feed consumption. Growth is more related to feed efficiency and feed conversion, i.e. the proportion of feed ingested that is effectively converted into biomass. I suggest reviewing.

We thank the reviewer for this important clarification. Although growth is indeed influenced by feed efficiency and feed conversion, in our study, the increased growth observed was primarily associated with higher feed intake, as no significant differences in FCR were found among treatments. This suggests that greater feed consumption drove growth improvements rather than changes in feed utilization efficiency.
Additionally, the diet provided to the different experimental groups was the same.

Line 408-410: To assess this gene, it might be interesting to do a postprandial sample instead of a post-fasting sample.

Thank the reviewer for the suggestion. We agree that assessing gene expression in a postprandial state could provide additional insights into feeding-related gene dynamics. In this study, we chose a post-fasting sampling to standardize metabolic conditions across groups, but future studies could certainly explore postprandial responses for a more detailed understanding.

Line 453-462: The conclusion cites previous work with zebrafish. Is this one of the aims of this research?

Thank the reviewer for the comment. The primary objective of this study was to evaluate the effects of selected synthetic flavorings on commercially relevant species, such as Seabass, based on preliminary research conducted on the zebrafish model. Testing the same flavorings previously tested on zebrafish provided essential data to support this work.

Round 2

Reviewer 1 Report

Comments and Suggestions for Authors

Ready for publication as is